biotechnology/microbiology/nanotechnology

graphene oxide, minimum inhibitory concentration, minimum bactericidal concentration, antibacterial activity, genotoxicity

**Author for correspondence:**
Abul Kalam Azad
e-mail: dakazad-btc@sust.edu

This article has been edited by the Royal Society of Chemistry, including the commissioning, peer review process and editorial aspects up to the point of acceptance.
†Joint first authors.

# Antibacterial activity of graphene oxide nanosheet against multidrug resistant superbugs isolated from infected patients

Md. Toasin Hossain Aunkor[1,†], Topu Raihan[1,†],
Shamsul H. Prodhan[1], H. S. C. Metselaar[2],
Syeda Umme Fahmida Malik[1,3] and Abul Kalam Azad[1]

[1]Department of Genetic Engineering and Biotechnology, Shahjalal University of Science and Technology, Sylhet 3114, Bangladesh
[2]Department of Mechanical Engineering, Faculty of Engineering, University of Malaya, 50603 Kuala Lumpur, W. Persekutuan Kuala Lumpur, Malaysia
[3]Department of Biochemistry, North East Medical College and Hospital, South Surma, Sylhet, Bangladesh

MTHA, 0000-0002-7202-5747; TR, 0000-0001-9406-6673;
SUFM, 0000-0001-6907-8226; AKA, 0000-0003-1918-3268

Graphene oxide (GO) is a derivative of graphene nanosheet which is the most promising material of the decade in biomedical research. In particular, it has been known as an antimicrobial nanomaterial with good biocompatibility. In this study, we have synthesized and characterize GO and checked its antimicrobial property against different Gram-negative and Gram-positive multidrug drug resistant (MDR) hospital superbugs grown in solid agar-based nutrient plates with and without human serum through the utilization of agar well diffusion method, live/dead fluorescent staining and genotoxicity analysis. No significant changes in antibacterial activity were found in these two different conditions. We also compare the bactericidal capability of GO with some commonly administered antibiotics and in all cases the degree of inhibition is found to be higher. The data presented here are novel and show that GO is an effective bactericidal agent against different superbugs and can be used as a future antibacterial agent.

## 1. Introduction

Bacterial infections and the contemporaneous development of resistance against antimicrobial agents have posed substantial

health risks in recent time. Excessive and improper administration of antibiotics in past decades played the major role in this crisis. In 2019, the World Health Organization (WHO) declared that antimicrobial resistance (AMR) will cause the death of 10 million people every year by 2050 with two-thirds of these deaths due to Gram-negative pathogens [1]. Notably, *Escherichia coli*, *Klebsiella pneumoniae*, *Pseudomonas aeruginosa*, *Proteus mirabilis*, *Serratia marcescens* and *Staphylococcus aureus* species are responsible for most of the clinical infections and thus form a major threat to human healthcare as a consequence of their multidrug resistance [2–5]. They employ different mechanisms to defend from multiple antibiotics, most notably: (i) by extracellular proteases assisted proteolytic degradation, (ii) sequestration caused by extracellular matrix, (iii) export of antibiotics by efflux pumps, (iv) steric hindrance generated by O-antigen of lipopolysaccharide (LPS), (v) enhanced rigidity by lipid A acylation, and (vi) electrostatic repulsion by a modified cell wall containing alanylated teichoic acids (TA), aminoacylated phosphatidyl glycerol (PG) and amine compound-added lipid A [6]. In order to save public health, novel antibacterial agents should be added to the arsenal to fight multidrug resistant (MDR) pathogens, especially before they develop new resistance mechanisms against current sensitive antibiotics [7]. In recent years, considerable research has been focused on the bottom-up development of nanoscale antibacterial materials as complementary or alternative agents owing to their enhanced antimicrobial activity and prolonged lifetime [8]. Graphene is the thinnest material on the earth. It is a plane sheet of $sp^2$ hybridized carbon, which is tightly confined into a honeycomb lattice [9]. Recently, the two-dimensional flat sheet of carbon has attracted the interest of bioscientists due to its extraordinary physical and chemical properties, particularly high surface-to-volume ratio and better biocompatibility [10–12]. Graphene oxide (GO) is produced from the chemical exfoliation of graphite and contains different oxidative functionalities, such as hydroxyl, carbonyl, epoxy and carboxylic groups, that enable its good water solubility. This phenomenon makes GO a potential candidate in human therapeutics. In many cases, GO can directly be used as produced or it can be reduced to obtain another promising product: reduced graphene oxide (rGO) [13]. While rGO also shows promising antimicrobial activity, it is not well soluble in polar solvents. The loss of polar functional groups during reduction causes the surface of rGO nanosheets to become hydrophobic and the strong van der Waals force among rGO nanosheets facilitates the aggregation of rGO particles. As a result, many black particles are observed to precipitate within the very first hour and this process continues until total sedimentation occurs within a few hours [14]. In order to get rid of this problem, rGO requires further chemical functionalization or modification for both *in vitro* and *in vivo* bio-applications [15]. Development of graphene-based antimicrobial nanocomposites is another highly focused area of current researchers which can be categorized in three wings: (i) graphene–metal nanocomposites (ii) graphene–metal oxide nanocomposites, and (iii) graphene–polymer nanocomposites [16–18]. Because the metal cations can directly attach to the oxygen groups on the GO surface by electrostatic interactions, graphene–metal nanocomposite development is getting extra importance in recent times. Thus, the deoxygenation occurs on the surface of GO nanosheets, forming stable metal-decorated graphene nanocomposites with antimicrobial activity, e.g. GO-Ag nanocomposites [19]. All of this additional processing of GO, i.e. the reduction, surface activation and composite formation, demand extra financial support to get pure biocompatible material of antimicrobial activity. By contrast, GO is highly stable in aqueous solution and has been shown to have a relatively greater bactericidal activity than that of rGO [20,21].

Herein, we synthesized GO using a modified Hummers' method improved by Shi and co-workers [22], because it requires a shorter oxidation time and it is free from the production of toxic gases and residual $Na^+$ and $NO_3^-$ while it produces very little under-oxidized hydrophobic carbon materials [23]. The product was characterized by different spectroscopic and microscopic techniques such as UV–visible spectroscopy, X-ray diffraction, Fourier transform infrared spectrometry, Raman spectroscopy, thermogravimetric analysis, energy dispersive X-ray spectroscopy, atomic force microscopy, scanning electron microscopy, field emission scanning electron microscopy and transmission electron microscopy. The antibacterial activity of well-dispersed and highly stable GO solution was then elucidated against five Gram-negative, *E. coli*, *K. pneumoniae*, *P. aeruginosa*, *P. mirabilis*, *S. marcescens*, and one Gram-positive, *S. aureus*, MDR superbugs directly isolated from clinical samples. Both serum-free and serum-containing media were used to determine and compare the minimum inhibitory concentration (MIC) and minimum bactericidal concentration (MBC) of GO in two different microenvironments. The results showed that GO could effectively inhibit and kill the aforementioned pathogens in both conditions in a concentration-dependent manner. The antibacterial activity of GO is compared to that of different commercial antibacterial discs usually prescribed by physicians. The *in vitro* DNA degradation of GO was checked and compared with the *in vivo*

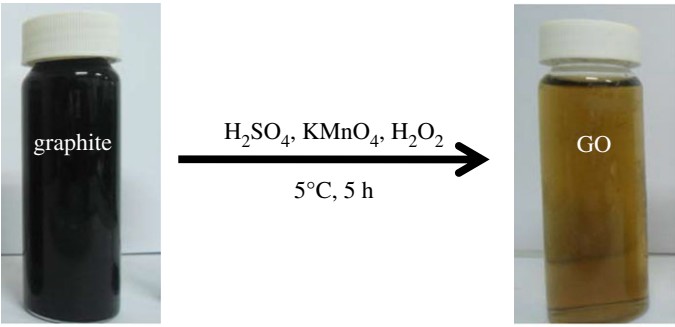

**Figure 1.** Chemical synthesis of GO by modified Hummers' method.

deterioration of bacterial DNA. The successful *in vivo* DNA destruction of GO suggests their cytoplasmic localization after puncturing the cell wall. The overall study provides new insights to arouse interest about GO as an alternative strategy for combating infections caused by multiple MDR pathogens.

# 2. Material and methods

## 2.1. Chemicals

Natural graphite flakes, concentrated sulfuric acid (98%), potassium permanganate, 30% hydrogen peroxide, concentrated hydrochloric acid (98%) and ultrapure water were purchased from Sigma-Aldrich, Malaysia. All of the aqueous solutions were prepared in deionized water. Analytical grade absolute ethanol, NaCl, peptone, yeast extract, beef extract, agar were purchased from Merck, Germany. Antibacterial discs: azithromycin (AZM), gentamycin (GEN), ciprofloxacin (CIP), cefixime (CFM), amoxicillin (AMX), cotrimoxazole (COT), imipenem (IPM) and ceftriaxone (CTR) were bought from Himedia Laboratories, India. 6X loading dye, 1 Kb plus DNA ladder and Presto Blue dye were purchased from Thermo Fisher Scientific. Human serum and bacterial strains collected from urine were obtained from North East Medical College Hospital, Sylhet, Bangladesh (https://www.nemc.edu.bd/pathology-department/) following all the ethical conditions and legislations approved by the ethical committee with the Ref. No. NEMC/Sylhet/287/2013 [24].

## 2.2. Synthesis of GO

Primarily, the beaker became heated after containing concentrated $H_2SO_4$ (80 ml), and therefore, it was placed in an ice water bath (0°C) to chill the excess temperature. Cautiously the graphite flake (3 g) was added to the beaker under mild agitation to get black colour graphite-acid solution. Subsequently, 5 g of $KMnO_4$ was mixed with the graphite-acid solution under gentle stirring at 5°C for 5 h. After that, the reaction mixture was poured in a 500 ml beaker and diluted by 350 ml of deionized water (DI). Then 30% $H_2O_2$ solution was added drop by drop until the mixture colour converted from black to bright yellow (figure 1). The bright yellow solution was kept overnight at room temperature. Finally, 100 ml of 5% HCl was added to dissociate exceeding manganese salt from the solution. Next, the brown-coloured GO dispersion was filtered out and washed repeatedly with ultrapure water till the pH of the material was 6.0. At the end, the purified product was placed in a dryer for 6 h at 70°C.

## 2.3. Spectroscopic and structural characterization of GO

The diluted aqueous solutions of GO were taken in crystal cuvette and placed in Hitachi (U-2001, Tokyo, Japan) for UV–vis measurement. The cuvettes were free from all kinds of finger print and spots to get an accurate result. DI water was used as reference for all the tests. Ringaku Mini Flex, 600, Japan was used to analyse XRD at 40 mA and 40 kV within the range of $2\theta = 5–50°$ using Copper K-α radiation. In Raman spectroscopy an inVia laser-Raman spectrometer (The Renishaw, UK) containing 514 argon-ion laser was run. Nicolet NEXUS 470 was applied to obtain FTIR spectra of graphite and GO. KBr was used to prepare sample pellets for the experiment. The TGA was conducted by Shimadzu TG 50 instrument, Japan at 10 °C min$^{-1}$ heating rate between 25 and 700°C temperature range in $N_2$ atmosphere. The SEM connected with EDX was obtained by Hitachi TM 3030 tabletop microscope, Japan. FESEM

**Table 1.** Zone of inhibition diameter generated by GO and commercial antibiotics.

| isolates | zone of inhibition (mm) | | | | | | | | |
|---|---|---|---|---|---|---|---|---|---|
| | CFM | CTR | COT | AZM | AMX | CIP | IMP | GEN | GO |
| E. coli | 0 | 12 | 0 | 19 | 0 | 27 | 0 | 26 | 39 |
| K. pneumoniae | 0 | 8 | 0 | 18 | 0 | 27 | 0 | 29 | 41 |
| S. aureus | 0 | 7 | 0 | 27 | 0 | 26 | 0 | 28 | 38 |
| P. aeruginosa | 0 | 12 | 0 | 20 | 0 | 27 | 0 | 25 | 38 |
| P. mirabilis | 0 | 8 | 0 | 29 | 0 | 27 | 0 | 25 | 27 |
| S. marcescens | 0 | 18 | 0 | 19 | 0 | 25 | 0 | 24 | 39 |

(AJSM-6700F with semi in-lens) was operated at 10 kV to get FESEM images at different magnifications. For the TEM, LEO-Libra 120 transmission electron microscope, Carl Zeiss AG, Germany was used. GO samples were sonicated for 20 min to dissolve well in water and one drop of the solution was cast on a fresh carbon-coated copper grid. The AFM sample was also prepared from the same solution by putting one drop of aqueous dispersion on $SiO_2/Si$ substrate. All of the above experiments were run several times for optimization.

## 2.4. Detection of antimicrobial activity of GO

To obtain a homogeneous suspension (1 µg µl$^{-1}$), 1 mg of synthesized GO powder was exfoliated into 1 ml DI water through ultrasonication at 20 kHz operating frequency and a maximum power of 500 W. During sonication, the amplitude setting was at 50% and 2 s ON and 2 s OFF pulses were applied [25,26]. The antibacterial activity of GO was conducted using the agar well diffusion method as described earlier [27]. The MDR bacterial isolates were cultured overnight in nutrient broth (0.5% NaCl, 0.5% peptone, 0.15% beef extract, 0.15% yeast extract and pH 6.8 ± 0.2) at 37°C and 120 r.p.m. in a rotary shaker. The pathogenic bacterial isolates were evenly cotton swabbed on nutrient agar (NA) according to McFarland standard 0.5 [28] and further a 5 mm well was created at the centre of each plate. A 20 µl of GO solution (1 µg µl$^{-1}$) was added in each well and incubated at 37°C. The clear zone of inhibition produced by the GO solution against the MDR bacterial isolates was measured and analysed according to the guidelines of the Clinical and Laboratory Standards Institute (CLSI) [29]. In order to compare the antibacterial activity of GO with that of the commercial antibiotics, commercial discs of GEN, AZM, IPM, COT, CFM, CIP, AMX and CTR were applied on the NA media cultured with the MDR bacterial isolates in accordance with the method aforementioned. The clear zones of inhibition found around the antibacterial discs were measured and analysed according to CLSI guidelines [29].

## 2.5. Determination of MIC and MBC of GO

The MIC and MBC of GO were determined against the pathogens mentioned in table 1 using the standard broth dilution method [30]. Initially, the pathogens were cultured in nutrient broth media and kept in a shaker incubator at 37°C overnight. A series of well-dispersed GO solutions (containing 1, 0.5, 0.25, 0.13, 0.065, 0.032, 0.016, 0.008 and 0.004 µg µl$^{-1}$) were prepared by twofold serial dilution using nutrient broth as diluent. Finally, 2 µl of each diluted bacterial isolate (McFarland standard 0.5) [28] was added to solutions of different GO concentrations and incubated at 37°C for 24 h. The lowest concentration of GO at which no visible bacterial growth was observed was termed MIC and the lowest concentration of GO at which all bacteria was completely destroyed was referred to as MBC. As GO solution is black it was very difficult to visualize the bacterial growth in the broth tube. For this reason, 200 µl cultures from every sample were spread on NA and incubated overnight at 37°C. The concentrations at which bacterial colonies were obtained indicating the MIC and that concentration of GO at which no bacterial growth was observed in the above condition is considered as the MBC of GO. In order to investigate the effect of serum on the MIC and MBC of the GO, a medium consisting of 80 ml sterile nutrient broth supplemented by 20 ml sterile blood (20% blood nutrient broth) was used.

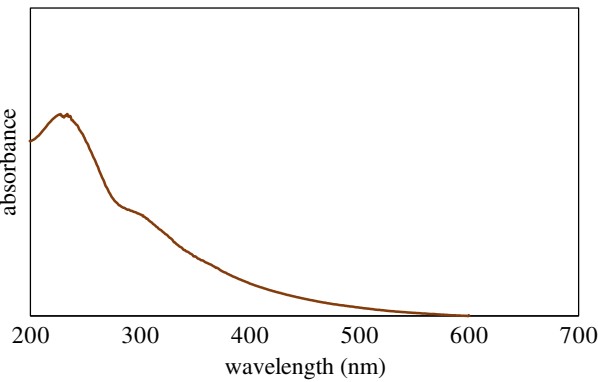

**Figure 2.** UV–vis spectrum of GO.

## 2.6. Cell viability in the presence of GO

The viability of GO-treated bacterial cells was investigated by Presto Blue cell viability assay in a U-bottom shaped microtitre plate. A 180 µl of fresh bacterial cultures were placed in the wells of the microtitre plate according to McFarland standard 0.5. The GO solutions of various concentrations (1–0.004 µg µl$^{-1}$) were added to the corresponding well containing bacterial suspensions and incubated at 37°C overnight. A 20 µl of Presto Blue was added in each well and kept in the incubator for 24 h. At the end, the microtitre plate was observed for visual colour change.

## 2.7. Analysis of genotoxicity of GO

*In vitro* genotoxic activity of GO was checked with minor modification of the method described previously [31]. *E. coli* was grown in nutrient broth overnight in a shaker incubator (120 r.p.m.) at 37°C and diluted by McFarland standard 0.5. Genomic DNA from *E. coli* was extracted as described previously [32]. A 5 µl of GO of different concentrations (concentrations: 1–0.004 µg µl$^{-1}$) was used to treat the DNA (10 µl) at 37°C for 2 h. A 10 µl of GO-treated DNA was mixed with 2 µl of 6X loading buffer and subjected to electrophoresis on 1% agarose gel. The gel was immersed in an ethidium bromide solution for 30 min and the DNA bands were visualized under UV transilluminator. The genotoxicity of GO was investigated *in vivo* by the slightly modified method described by Li and co-workers [33]. The *E. coli* was grown in the presence of different concentrations of GO solutions (concentration: 1–0.004 µg µl$^{-1}$). A 1.5 ml of bacterial suspension was taken for genomic DNA extraction [32]. The isolated DNA (10 µl) was subjected to electrophoresis on 1% agarose gel and visualized by the method aforementioned.

# 3. Results and discussion

## 3.1. Spectroscopic characterizations of GO

UV–vis absorption spectroscopy is generally used to assure the presence of oxygen-carrying groups. The UV–vis spectrum shows an absorption peak at 231 nm wavelength, which demonstrates the π–π$^*$ transition of skeletal C=C bonds (figure 2). Additionally, the spectrum also represents a tiny shoulder at around 300 nm ascribed to the n–π$^*$ transition of carbonyl groups (COOH) that is consistent with the previous reports [34,35]. Zhang and co-workers [36] reported that the molar absorbance of GO is size dependent. They represented that the small (hydrodynamic diameter 1 µm), medium (1.5 µm) and large (2.2 µm) sized GO particles display absorbance band at 210, 230 and 250 nm, respectively. Our GO samples displayed absorbance peak at 231 nm can be referred to as medium-sized sheets. This was confirmed by AFM measurement at described in the next section. The gap between two layers is an important parameter to study the structural information of GO. The XRD helps to assess the crystalline structure of GO nanosheets. The XRD profile of pristine graphite and GO are shown in figure 3. In pristine graphite a strong and sharp 002 diffraction peak can be seen at $2\theta = 26.49°$ corresponding to a d-spacing 3.35 Å (figure 3*a*). After oxidation of graphite to GO, this characteristic peak (002) has been shifted to a lower angle [37–40]. In the current study, it has appeared at $2\theta = 10.43°$ with a d-spacing of 8.47 Å

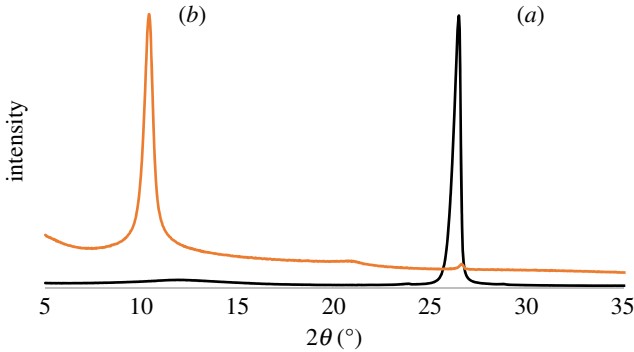

**Figure 3.** XRD patterns of (*a*) graphite and (*b*) GO.

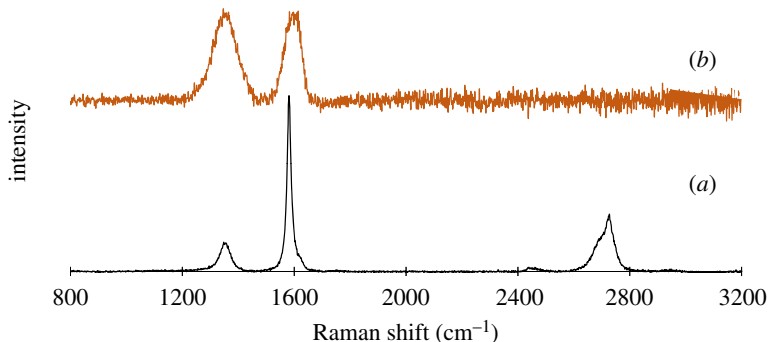

**Figure 4.** Raman spectroscopy of (*a*) graphite and (*b*) GO.

(figure 3*b*). The increase of d-spacing from 3.35 to 8.47 Å confirms the addition of oxygen functional groups in between the graphene sheets of pristine graphite to assemble restacked GO. It also indicates the creation of new sp$^3$ bonding, responsible for structural defects or atomic scale roughness [41]. Raman spectroscopy is a chief technique to characterize GO nanosheets. It is used to analyse the order/disorder crystal structure of carbon-related materials. Generally, the Raman spectra of graphene-related materials show two original peaks within the range of 1300–1700 cm$^{-1}$ called D peak and G peak. The characteristic D peak (for disorder) is the result of a double resonance that is not Raman active in defect-free graphite. Secondly, the G peak (for graphite) is the result of first-order scattering of E$_{2g}$ phonons from sp$^2$ bonded carbon. The 2D peak, an overtone of the D peak, is active even in defect-free graphite [42]. In the current study, a sharp graphitic G peak at 1581 cm$^{-1}$ and a D peak at 1353 cm$^{-1}$ are observed (figure 4*a*), indicating the nano-crystalline nature of the used graphite [43]. Here, the 2D band at 2725 cm$^{-1}$ evolves the number of graphene layers in the bulk graphite [44]. After the treatments of oxidizing agents, the Raman spectrum of as-prepared GO shows the D peak and G peak at 1350 and 1614 cm$^{-1}$, respectively (figure 4*b*). Herein the G band is broadened and blue-shifted to higher wavelength, expressing size degeneration of the in-plane sp$^2$ domains as a result of extensive oxidation [45]. The intensity ratio, $I_D/I_G$ is close to one, which is typical for GO prepared by Hummers' method [46]. FTIR is a characterization technique, which gives information about functional groups in all types of materials including nanoscale materials. The FTIR spectra of pristine graphite as exhibited in figure 5*a* contains only two characteristic peaks at 1610 cm$^{-1}$ (C=C stretching vibration) and 3450 cm$^{-1}$ (O–H stretching vibration) attributed to aromatic C=C bonding and adsorbed water molecules [47]. In GO, the emergence of various intense bands at around 1052, 1219, 1386 and 1706 cm$^{-1}$ are related to the alkoxy group (C–O stretching), epoxy ring (C–O–C), OH deformation and carbonyl group (–C=O), respectively (figure 5*b*). Importantly, another broad band at approximately 3395 cm$^{-1}$ is liable for OH stretching. All these peaks confirm the penetration of the above oxygen moieties over the graphene sheet [48]. Notably, among the aforementioned oxygen groups, the hydroxyl and epoxy groups are predominantly located at the basal plane while the carboxyl groups are found at the edge of the sheet [49]. This occurrence also alludes defect formation by sp$^2$ character destruction owing to expansive oxidation. Significantly this conjoining of enormous oxygen-containing groups makes GO hydrophilic and highly stable in

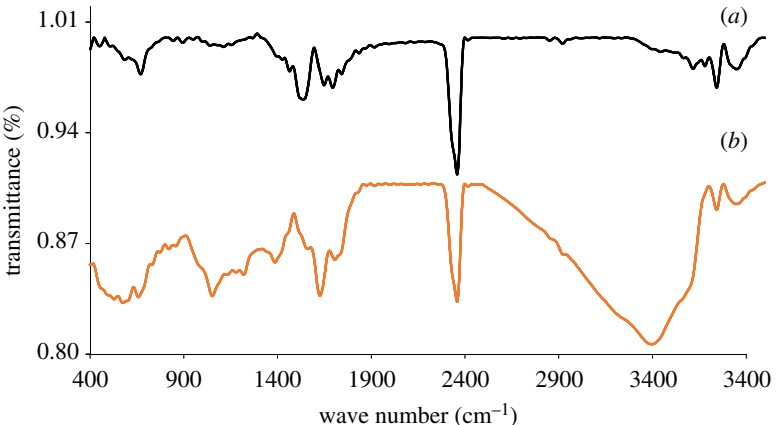

**Figure 5.** FTIR spectroscopy of (*a*) graphite and (*b*) GO.

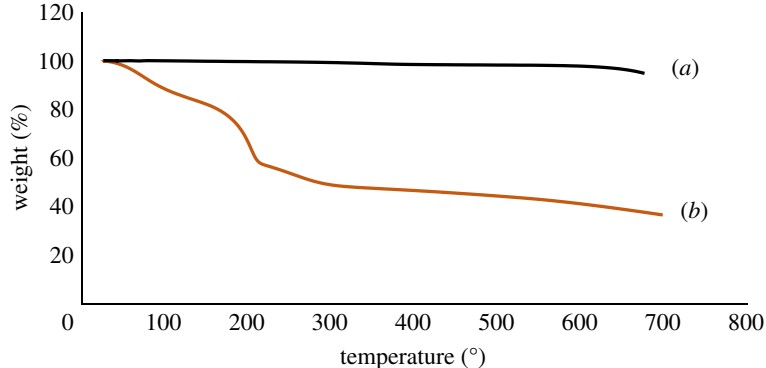

**Figure 6.** TGA plots of (*a*) graphite and (*b*) GO.

polar solvents [50]. Additionally, the appearance of one strong peak at 2350 cm$^{-1}$ for both graphite and GO is attributed to the O=C=O stretching of gaseous $CO_2$ absorbed by the material surface [51,52]. The thermal stability of graphite and GO were investigated by TGA. Graphite shows good thermal stability even at 700° C as shown in figure 6*a*. By contrast, GO is thermally unstable and begins to lose weight just after heating [53]. The curve of GO displayed two significant stages of mass loss (figure 6*b*). The first stage displayed about 9% mass loss around 100°C which states the evaporation water molecules absorbed by the sheets. It also indicates that around 9% of water residues could be entrapped between the GO sheets [54,55]. The second stage displayed at about 220°C with 43% mass losses. It is owing to the decomposition of labile oxygen moieties by heat. This weight reduction continued with temperature and finally showed about 60% mass losses at 600°C. It is related to the burning of the carbon skeleton and pyrolysis of subsisting moieties. Herein the residual weight of GO is nearly 36% at 700°C. These results suggest that the thermal behaviour of graphite has been changed upon conversion to GO [56,57]. It is important to study the material purity as well as elemental composition analysis in nanomaterial synthesis where energy dispersive X-ray spectrometry (EDX) plays the vital role. Electronic supplementary material, figure S1a shows 98% atomic weight percentage of carbon in pristine graphite with 2% remaining aerial or atmospheric oxygen bound with the sheet surface through weak van der Waals interactions. By contrast, electronic supplementary material figure, S1b shows 55.8% and 38.9% atomic weight percentage of carbon and oxygen, respectively [58]. This rising of oxygen percentage in the atomic level refers to the successful oxidation of graphite to GO. Trace amounts of sulfur (atomic weight percentage 2.6%) are also found in the sample, which is from the $H_2SO_4$ used in synthesis.

## 3.2. Microscopic characterizations of GO

The SEM images of graphite and synthesized GO were obtained to view the plane morphology of the materials. Morphologically, natural graphite powder has a flaky appearence (electronic supplementary material, figure S2a), which is due to the strong sp$^2$ carbon–carbon bonding. Contrariwise, the surface

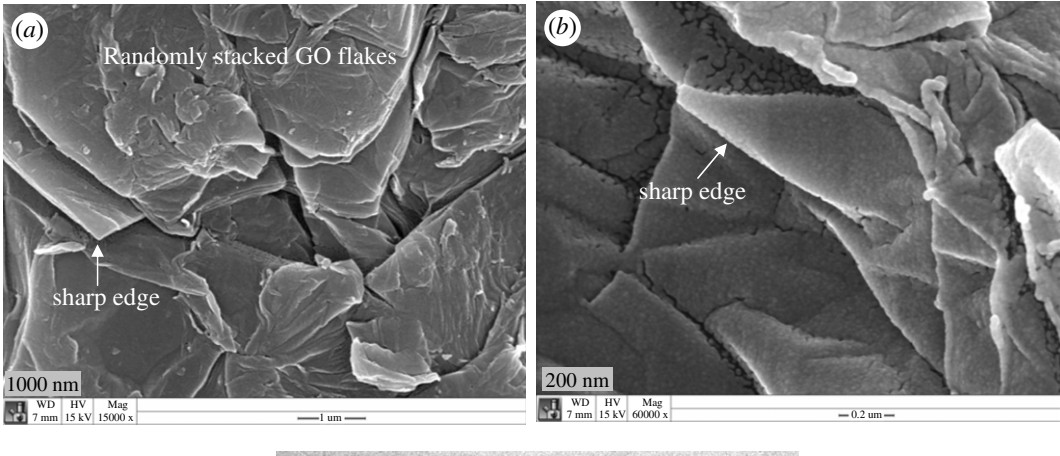

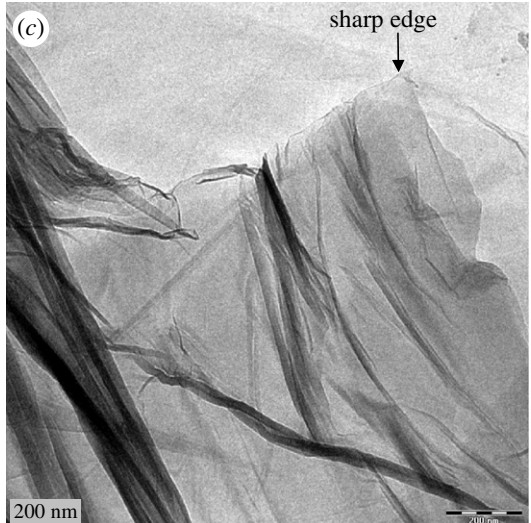

**Figure 7.** FESEM of (*a*) GO stacks, (*b*) sharp edge of GO and (*c*) TEM of GO nanosheet.

of GO exhibits carpet appearance as is seen in electronic supplementary material, figure S2b. It may indicate the localization of hydroxyl and carboxyl groups as well as residual water molecules to carbon surface due to extensive oxidation [59]. Figure 7*a* displays the FESEM image of GO nanosheets. Here, the crystal and shrivelling appearance of GO were observed where the sheets were slightly amalgamated with each other. Usually, it is known as overlapping or random orientation leading to develop agglomerate in solid state [60,61]. Figure 7*b* shows the higher magnification FESEM of GO. From the image, it can be seen that the edges of the sheets are sharp with uniform inner surface morphology. The sharp edge is further confirmed by TEM, as depicted in figure 7*c*. In addition to this, the appearance is thin, transparent and smooth with small wrinkles, which is the intrinsic nature of GO [62–64]. AFM is another sophisticated tool to investigate the morphology and topography of graphene or its derivative materials. It is able to measure the height profiles and surface area of a single GO sheet. Therefore, it also allows identifying the successful synthesis of GO as well as helps to count the number of definite layers in the sample. Figure 8*a* displays AFM image of GO run in tapping mode. However, it is well documented that the van der Waals layer spacing of bulk graphite is around 0.34 nm. It also looks automatically flat and the honeycomb surface becomes apparent under AFM imaging [45]. After oxidation, the thickness of newly synthesized GO is increased reconfirming the accumulation of covalently bonded oxygen moieties. This adjoining of oxygen lapses the $sp^2$ hybridized carbons above and beneath the original sheets of graphene. In this study, the average thickness of GO nanosheet is found around 1.20 nm after ultrasonication assisted exfoliation in water; that is less than two times the layer spacing found by XRD, suggesting the presence of single-layer exfoliated sheets having sharp edges. Figure 8*b* displays the lateral size of single GO nanosheet at 1.6 µm; although the lateral size of most of the GO sheets in our sample is obtained within the range of 1.5–2 µm suggesting the formation of medium-sized leaf-like exfoliated GO [65,66]. Here some particle-like features can be also seen on the surface of the sheets. These particle-like features are formed of residual carbonaceous materials attached to defect sites of the

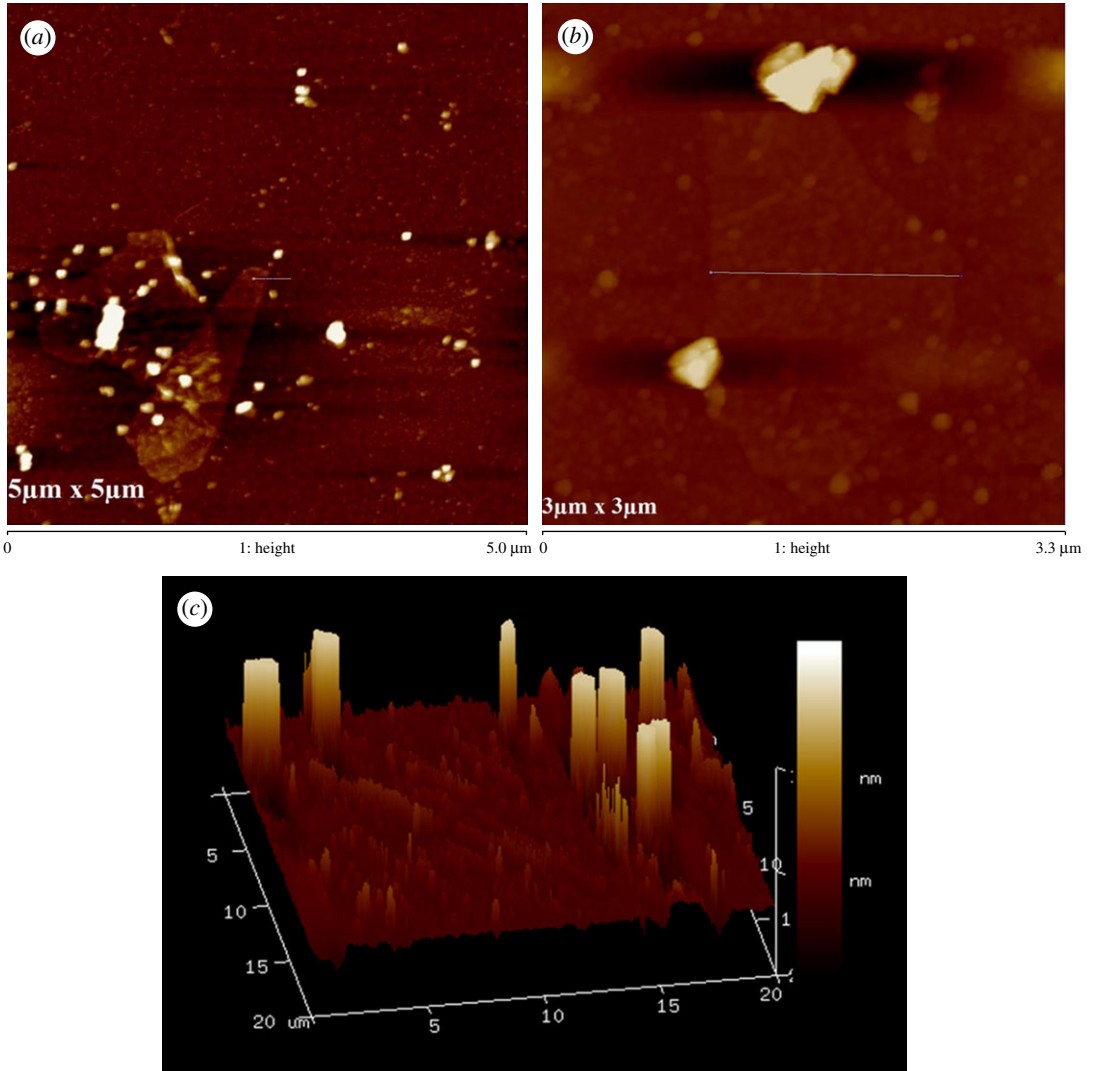

**Figure 8.** Tapping mode AFM measurement of (*a*) GO (5 × 5 µ scan area, (*b*) lateral dimension (3 × 3 µ scan area) and (*c*) three-dimensional topography of GO deposited on to SiO₂/Si substrate.

sheets, as reported previously [67–69]. Furthermore, the three-dimensional AFM image (figure 8*c*) of a diluted GO droplet deposited on a Si substrate reveals a jagged surface. The corresponding height profiles of figure 8*a,b* are available in the electronic supplementary material, figure S3.

## 3.3. Antibacterial activity

GO showed strong antibacterial activity against the Gram-positive and Gram-negative MDR pathogens tested, by forming zone of inhibition in an agar well diffusion assay (figure 9 and table 1). The antibacterial activity of GO was compared with that of the commercial antibiotics GEN, AZM, IMP, COT, CFM, CIP, AMX and CTR (figure 9). All the pathogens used in the study are sensitive to GO but they are totally resistant to CFM, COT, AMX and IPM. GEN, AZM, CTR and CIP produced a zone of inhibition in the range of 7–29 mm, whereas, GO generated a clear zone of inhibition in the range of 27–41 mm, indicating that GO is an attractive antibacterial agent. The FESEM, TEM and AFM studies revealed that the synthesized GO had sharp edges which might act as 'Nano knives'. Upon direct contact with the bacterial outer surface, this may puncture the cell wall and the membrane [70,71]. This may lead to lipid injuries (scheme 1), where large amounts of phospholipids may be extracted from the microbes by further van der Waals and hydrophobic bond formation with the charged groups of GO [72]. As a result, the pathogens may suffer from perturbation of membrane integrity which can disable many essential functions, such as respiration, materials transport, osmotic balance and energy transduction [67,70,73,74]. If the surface of the bacterial cell is surrounded by sufficient

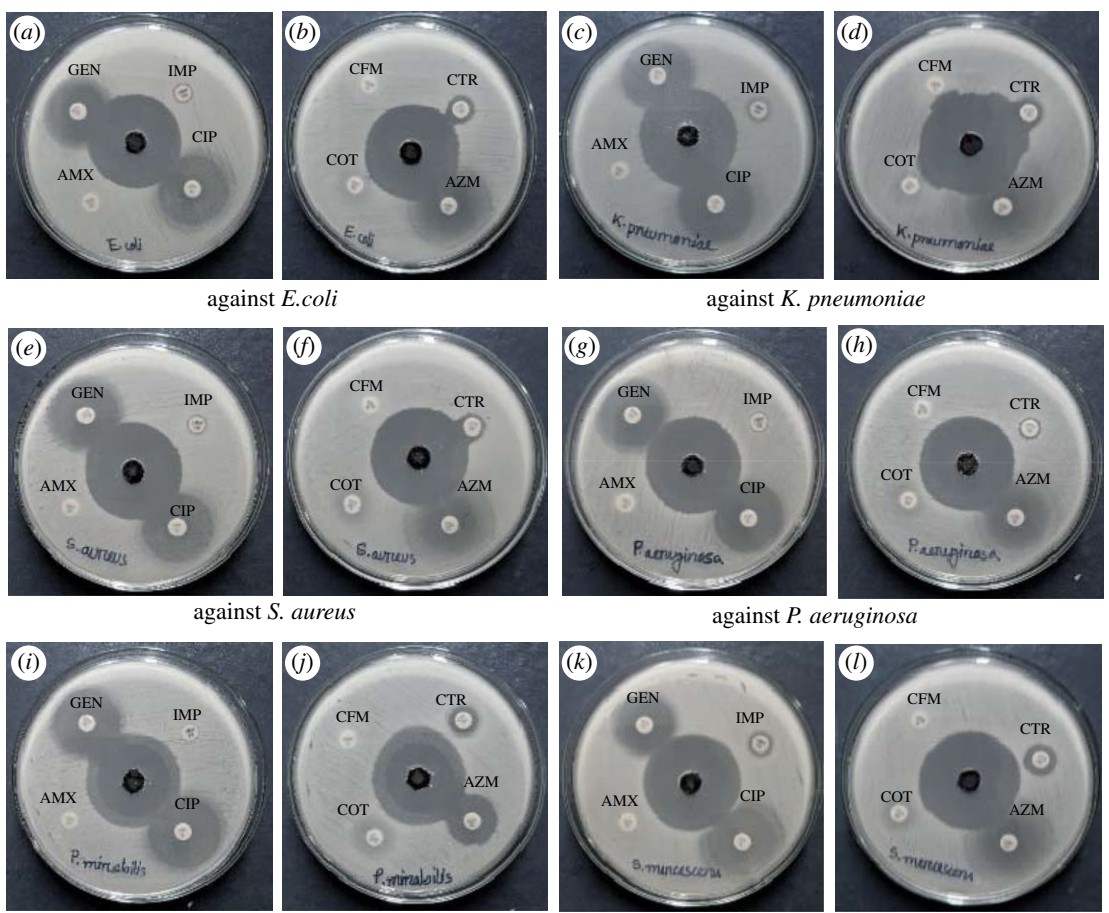

**Figure 9.** *In vitro* antibacterial activity of GO and commercial antibiotics against *E. coli* (*a*), (*b*); *K. pneumoniae* (*c*), (*d*); *S. aureus* (*e*), (*f*); *P. aeruginosa* (*g*), (*h*); *P. mirabilis* (*i*), (*j*); and *S. marcescens* (*k*), (*l*).

GO sheets, the cell becomes biologically inactive and finally dies, leading to the formation of a clear zone of inhibition in the culture plate (scheme 1). Secondly, cell entrapment might be another potential mechanism for the antibacterial activity of GO. The bacterial cells may be trapped by GO sheets upon their contact. The trapped bacteria might be detached from the external microenvironment and their access to nutrients might be restricted causing growth inhibition. Herein, the size of GO may have a significant influence on cell entrapment, with sheets of large lateral size showing better inhibition [65]. Based on UV–vis and AFM data, the lateral size of the GO in the present study ranged from 1.5 µm to 2.0 µm which is similar to the size of rod-shaped (1–2 µm) bacteria, *E. coli*, *K. pneumoniae*, *P. aeruginosa*, *P. mirabilis* and *S. marcescens*, and slightly bigger than the round-shaped (up to 1 µm) bacteria, *S. aureus*, alluding to the possibility of entrapment. Thus, it can be assumed that the GO nanosheets are able to kill bacteria by both entrapment and edge-mediated cutting, although it requires more study of the samples produced in our laboratory.

## 3.4. MIC and MBC of GO

The MIC and MBC were determined in the absence and presence of serum in the broth. The purpose of the MIC and MBC analysis in both microenvironments was to distinguish the killing performance of GO in artificial and blood containing media, because after injection into the bloodstream, the activity of GO may be modified by adsorption of blood proteins or other biomolecules. The MIC of GO in the absence of serum was 0.065 µg µl$^{-1}$ against *E. coli*, *K. pneumoniae*, *P. mirabilis* and *S. aureus* while it was 0.032 µg µl$^{-1}$ against *P. aeruginosa* and *S. marcescens* (electronic supplementary material, table S1). The MIC of GO did not change against the bacteria aforementioned in the presence of serum in the broth. One reason for subsequent colony formation on the solid media could be the repair of membrane damage in the new environment provided by the NA. The reason of colony formation from the broth with MIC of GO in

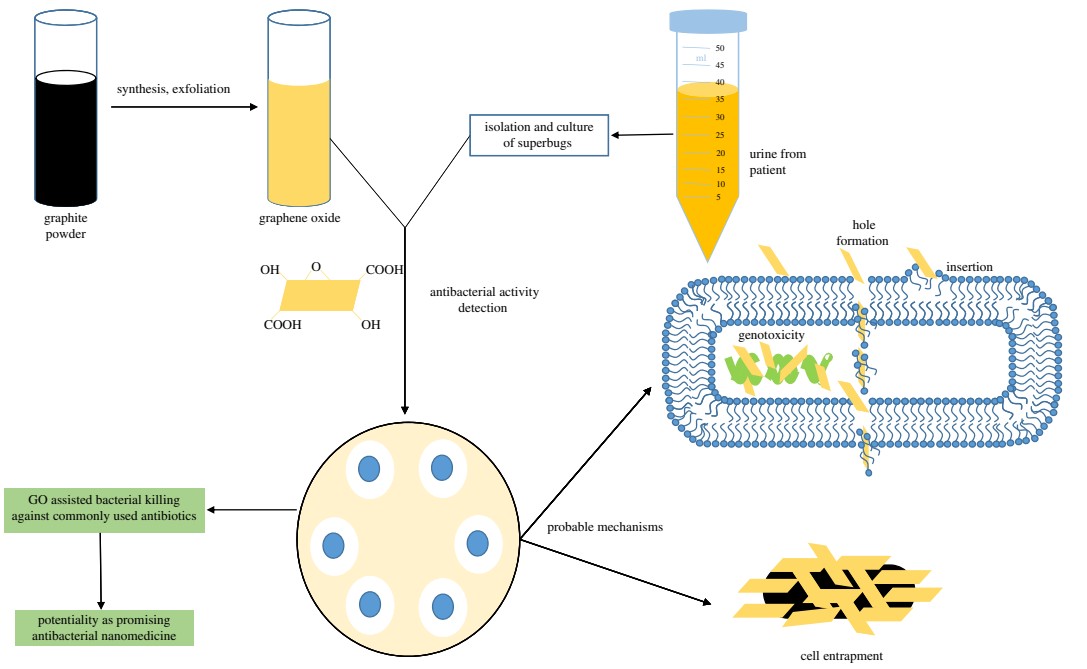

**Scheme 1.** Bactericidal activity of GO against clinical superbugs.

the presence of serum could be the non-specific binding of blood macromolecules with the GO. It may produce corona and reduce the available surface area, which in turn hinders the cutting effect of bacterial membrane or extraction of phospholipids to create membrane pores. In addition, an unfavourable geometric barrier or steric effect may emerge, most specifically an electrostatic interaction between peripheral residues of GO containing biocorona and the head group of membrane lipids [75,76]. By contrast, with the increment of GO concentration, the degree of damage is excessive and overwhelms the capability of the repair system. Thus, bacterial strains cannot regenerate the ruptured membrane and finally fully perish at their corresponding MBC in the absence and presence of serum [77]. The MBC of GO against *P. aeruginosa* and *S. marcescens* was 0.065 µg µl$^{-1}$ and that against other bacteria tested was 0.13 µg µl$^{-1}$ (electronic supplementary material, table S1). This finding was further confirmed by Presto Blue cell viability assay as shown in figure 10a. Presto Blue is a cell permeable resazurin-based weakly fluorescent blue colour dye which uses the reducing power of living cells to measure cell proliferation. In principle, during incubation of living cells with Presto Blue, resazurin can diffuse into the cell where it can be reduced to resorufin by the action of different cytoplasmic redox enzymes particularly NADPH-dependent dehydrogenase (figure 10b). Resorufin is red in colour and emits fluorescence at 590 nm, indicating the presence of viable cells. This conversion is proportional to the number of metabolically active/viable cells providing a quantitative measurement. By contrast, upon co-culture with bacteria, the appearance of dye in its original colour (blue) confirms cell death. In this assay, wells containing 1, 0.5, 0.25, 0.13, 0.065 µg µl$^{-1}$ of GO against *E. coli*, *K. pneumoniae*, *P mirabilis*, *S. aureus* and 1, 0.5.0.25, 0.13, 0.065, 0.032 µg µl$^{-1}$ against *P. aeruginosa* and *S. marcescens* appeared blue indicating the cytotoxic behaviour of the nanomaterial. To reconfirm MIC among these concentrations, all of the contents from blue wells were directly plated on NA. Bacterial colonies of *E. coli*, *K. pneumoniae*, *P. mirabilis*, *S. aureus* were found at 0.065 µg µl$^{-1}$ of GO and those of *P. aeruginosa* and *S. marcescens* were found at 0.032 µg µl$^{-1}$ of GO, which supported our previous result obtained by the broth dilution method. On the other hand, the MBC was recorded unchanged because there was no colony growth with the content of the well.

## 3.5. *In vitro* and *in vivo* genotoxic effect

Genotoxicity describes the property of a material that damages the genetic information within a cell. The objective of this study is to bring out the adverse effect of GO on bacterial DNA, which controls the resistance mechanism against biocides. The intensity of a DNA band in agarose gel has a positive relation with the DNA concentration, but for GO-treated DNA the band intensity is inversely proportional to the concentration of GO [33]. The fine band in lanes 1, 2 and 3 (figure 11a) suggests

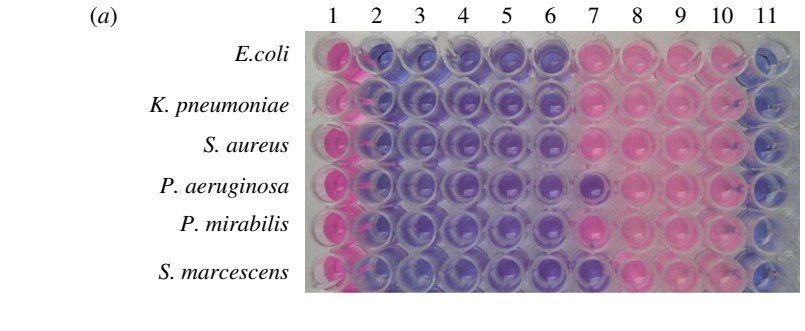

(a)

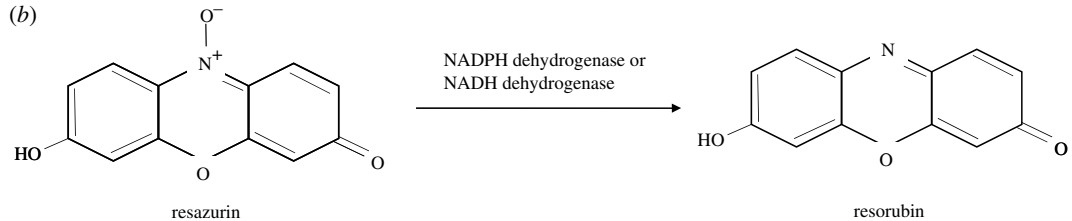

(b)

**Figure 10.** Presto Blue cell viability assay. Column 1 contains only bacterial culture; columns 2–10 contain GO solution of 1, 0.5, 0.25, 0.13, 0.065, 0.032, 0.016, 0.008 and 0.004 µg µl$^{-1}$, respectively, and pathogenic bacterial culture; column 11 contains only broth media (a). Bioreduction of resazurin to resorufin (b).

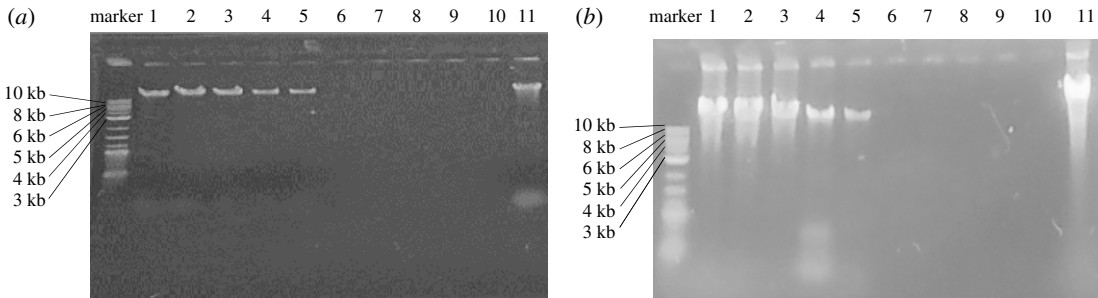

**Figure 11.** Electrophoresis analysis of DNA extracted from *E. coli* treated with various concentrations of GO. Lane 11 contains untreated bacterial DNA; lane 10 contains nothing. Lanes 1–9 contains DNA treated with various concentrations of GO (1, 0.5, 0.25, 0.13, 0.065, 0.032, 0.016, 0.008 and 0.004 µg µl$^{-1}$) in *in vitro* condition (a). Lanes 2–9 contains DNA treated with various concentrations of GO (1, 0.5, 0.25, 0.13, 0.065, 0.032, 0.016, 0.008 and 0.004 µg µl$^{-1}$) in *in vivo* condition (b).

that the used GO concentrations have no genotoxic effect against the extracted DNA. But band intensity is reduced at the concentrations loaded in lanes 4 and 5 indicating moderate DNA degradation. From lanes 6–9 the invisibility of band is an indication of successful DNA damage. Two possible mechanisms are involved in this band disappearance: (i) ethidium bromide may bind with GO nanosheets and lose its fluorescence due to electron transfer from the structural charged groups to GO surface functionalities via different electrostatic attraction [78], and (ii) the positively charged ethidium 3, 8-exocyclic amine groups of ethidium bromide bind with negatively charged DNA phosphate through hydrogen bond formation and emit 600 nm light upon UV excitation. But surrounding GO residues may attach with the DNA by π–π stacking and quench the dye by fluorescence resonance energy transfer (FRET) [79–81]. The concentration of the DNA band for treated cells was decreased in a dose-dependent manner. Conversely, DNA extracted from GO-treated bacterial cells displays clear bands from lanes 1–3 (figure 11b) indicating that the bacteria were able to grow normally with no effect of GO to their genetic material. Interestingly, for lanes 4 and 5 the band intensity was shortened, indicating a partial genotoxic effect of the applied amounts. However, no band was observed in lanes 6–9 suggesting that the biological structure of DNA was completely broken by GO *in vivo*.

# 4. Conclusion

In conclusion, GO nanosheets were synthesized using a modified Hummers' method and characterized. The antimicrobial activity of GO nanosheets was studied on both serum-free and serum-containing media against different life-threatening MDR superbugs namely *E. coli*, *K. pneumoniae*, *P. aeruginosa*, *P. mirabilis*, *S. marcescens* and *S. aureus* which were directly collected from affected persons. It was exposed that the antimicrobial effect of GO nanosheets is prominent and the blood components do not reduce its activity on media plates. The determination of MIC revealed the last line concentration of its fruitful administration as drug. With the increased concentration (MBC), GO nanosheets work more effectively to restrain the viability of these pathogens. We conjecture that this occurs by puncturing cell walls and disbanding the bacterial genome as well as isolating the bacteria from the growth media. Moreover, we compared the bactericidal activity of GO with GEN, AZM, IPM, COT, CFM, CIP, AMX and CTR, implying its potency to use as a new antibiotic. Significantly, the strains we used are also considered as biofilm formers where medical implants are ideal surfaces for their colonization. Notably, biofilm formation is a complex biological process and is regulated by different physico-chemical and environmental factors at the interface of the cell and surface of the substrate, facilitating cell adhesion, microcolony formation, and the release of the exopolymeric matrix as adhesive glue to protect the embedded cells from UV light, radiation, pH changes, osmotic shock, drying and most significantly shield from antibiotic or biocide attack. This biofilm can be built in both short-term devices such as urinary catheters, as well as long-term implants like artificial joints and dental sealants [82–84]. It is projected that more than $1.62 billion/year financial burdens will be incurred by 2020 on the healthcare system due to this implant-related infection [85]. Our study opens another way of research to develop a GO-based antibiofilm surface coatings technology for healthcare and biomedical applications.

Ethics. Human whole blood samples were obtained from North East Medical College Hospital, Sylhet, Bangladesh. The institutional ethical committee reviewed and approved the transfer of blood to our institute (Ref. no.: NEMC/Sylhet/291/2013).

Data accessibility. The spectroscopic raw data of UV–vis, XRD, Raman, FTIR and TGA is available in the Dryad Digital Repository: https://dx.doi.org/10.5061/dryad.ns1rn8ppr [86].

Authors' contributions. Md.T.H.A. and T.R. equally contributed to perform experiments, data analysis and writing the draft manuscript. S.H.P. and H.S.C.M. participated in manuscript revision. S.U.F.M. collected blood samples and participated in some experiments. A.K.A. designed experiments, analysed and interpreted the data with editing manuscript. All authors have approved the final article.

Competing interests. The authors declared no potential conflicts of interest with respect to the research, authorship, and/or publication of this article.

Funding. There was no external fund for the study.

Acknowledgements. This work was partially supported by a grant in aid from the Research Centre, Shahjalal University of Science and Technology, Sylhet, Bangladesh (no. LS/2019/1/13).

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
