## [Reviewer comments · Royal Society Open Science]

Review History

RSOS-200640.R0 (Original submission)

Review form: Reviewer 1 (Zakaria Mia)

Is the manuscript scientifically sound in its present form?

Yes

Are the interpretations and conclusions justified by the results?

Yes

Is the language acceptable?

Yes

Do you have any ethical concerns with this paper?

No

Have you any concerns about statistical analyses in this paper?

No

Recommendation?

Accept with minor revision (please list in comments)

Comments to the Author(s)

10 million people every year by 2050. with (page 3, row 35)

----- [Recommendation-1: Please omit the dot (.) after 2050]/

350 ml of deionized water (DI). Than 30% (page 5, row 118)

----- [Recommendation-2: Please write 'Then' instead of 'Than']/

the pH of the material was at 6. (page 6, row 122)

----- [Recommendation-3: write the value as '6.0' instead of 6 only, though the value same]/

20 μ l of prestoblue was added in each well and (page 8, row 186)

----- [Recommendation-4: start with words, not with numerical figure]/

used to treat the DNA (10 μ l) at 37°C for 2 h. (page 8, row 195)

----- [Recommendation-5: if possible, please mention the concentration of DNA (optional)]/

10 μ l of GO treated DNA was mixed with 2 μ l of.... (page 8, row 195)

----- [Recommendation-6: start with words, not with numerical figure]/

Thus it can be assumed that the (page 12, row 331)

----- [Recommendation-7: Please put a comma just after 'Thus']/

Thus bacterial strains cannot regenerate the ruptured membrane and (page 13, row 349)

----- [Recommendation-8: Please put a comma just after 'Thus']/

For fig. 3 and 4 (Page 30, 31)

----- [Recommendation-9: if appropriate, Label the axis (X- and Y)]/

For fig. 6 (Page 33)

----- [Recommendation-10: if appropriate, Label the axis (X- and Y)]

Review form: Reviewer 2

Is the manuscript scientifically sound in its present form?

Yes

Are the interpretations and conclusions justified by the results?

Yes

Is the language acceptable?

Yes

Do you have any ethical concerns with this paper?

No

Have you any concerns about statistical analyses in this paper?

No

Recommendation?

Accept with minor revision (please list in comments)

Comments to the Author(s)

See the attached file (Appendix A).

Decision letter (RSOS-200640.R0)

Dear Dr Azad:

Title: Antibacterial Activity of Graphene Oxide Nanosheet against Multi Drug Resistant Superbugs Isolated from Infected Patients
Manuscript ID: RSOS-200640

The editor assigned to your manuscript has now received comments from reviewers. We would like you to revise your paper in accordance with the referee and Subject Editor suggestions which can be found below (not including confidential reports to the Editor). Please note this decision does not guarantee eventual acceptance.

Please submit your revised paper before 27-Jun-2020. Please note that the revision deadline will expire at 00.00am on this date. If we do not hear from you within this time then it will be assumed that the paper has been withdrawn. In exceptional circumstances, extensions may be possible if agreed with the Editorial Office in advance. We do not allow multiple rounds of revision so we urge you to make every effort to fully address all of the comments at this stage. If deemed necessary by the Editors, your manuscript will be sent back to one or more of the original reviewers for assessment. If the original reviewers are not available we may invite new reviewers.

RSC Associate Editor:
Comments to the Author:
(There are no comments.)

RSC Subject Editor:
Comments to the Author:
(There are no comments.)

Reviewers' Comments to Author:
Reviewer: 1

Comments to the Author(s)
10 million people every year by 2050. with (page 3, row 35)
----- [Recommendation-1: Please omit the dot (.) after 2050]/
350 ml of deionized water (DI). Than 30% (page 5, row 118)
----- [Recommendation-2: Please write 'Then' instead of 'Than']/
the pH of the material was at 6. (page 6, row 122)
----- [Recommendation-3: write the value as '6.0' instead of 6 only, though the value same]/
20 µl of prestoblue was added in each well and (page 8, row 186)
----- [Recommendation-4: start with words, not with numerical figure]/
used to treat the DNA (10 µl) at 37°C for 2 h. (page 8, row 195)
----- [Recommendation-5: if possible, please mention the concentration of DNA (optional)]/
10 µl of GO treated DNA was mixed with 2 µl of... (page 8, row 195)
----- [Recommendation-6: start with words, not with numerical figure]/
Thus it can be assumed that the (page 12, row 331)
----- [Recommendation-7: Please put a comma just after 'Thus']/
Thus bacterial strains cannot regenerate the ruptured membrane and (page 13,row 349)
----- [Recommendation-8: Please put a comma just after 'Thus']/
For fig. 3 and 4 (Page 30, 31)
----- [Recommendation-9: if appropriate, Label the axis (X- and Y)]/
For fig. 6 (Page 33)
----- [Recommendation-10: if appropriate, Label the axis (X- and Y)]

Reviewer: 2

Comments to the Author(s)

See the attached file

Author's Response to Decision Letter for (RSOS-200640.R0)

See Appendices B & C.

Decision letter (RSOS-200640.R1)

Dear Dr Azad:

Title: Antibacterial Activity of Graphene Oxide Nanosheet against Multi Drug Resistant Superbugs Isolated from Infected Patients
Manuscript ID: RSOS-200640.R1

It is a pleasure to accept your manuscript in its current form for publication in Royal Society Open Science. The chemistry content of Royal Society Open Science is published in collaboration with the Royal Society of Chemistry.

RSC Associate Editor
Comments to the Author:
(There are no comments.)

Reviewer(s)' Comments to Author:

Appendix A

In this work, the authors synthesized graphene oxide (GO) using a modified Hummers' method and characterized using a variety of materials characterization techniques such as UV-Vis, FTIR, Raman, SEM, EDX, TEM, AFM, XRD, and TGA. The antibacterial activity of GO was then evaluated against five gram-negative and one gram-positive multi drug resistant superbugs grown in two different microenvironments (i.e., serum-free and serum-containing media). The authors also compared the antibacterial activity of GO with some common antibiotics. This work is suitable for publication in RSOS, and it can be considered for publication after addressing the following comments and questions:

- 1. Abstract (Lines 23-26):** The authors stated, "*The main focus of the antimicrobial activity analysis is... proteins or other biomolecules.*"; this sentence is not necessary in Abstract and can be moved to Antibacterial Activity discussion section.
- 2. Introduction:**
 - Lines 48-50:** Add proper references at the end of this sentence: *In recent years, considerable research has been focused on the bottom-up development of ... and prolonged lifetime.*^X
 - Line 78:** The authors stated, "*Herein, we synthesized GO using a modified Hummers' method ...*"; which modification of Hummers' method exactly? add a proper reference.
 - Line 79:** Correct NO₃ to NO₃⁻
 - Line 98:** Scheme 1 is mentioned here, but not discussed! I recommend the authors to move Scheme 1 to the beginning of Results and Discussion section and discuss it there.
- 3. Materials and Methods:**
 - Lines 114-123:** The authors provided a qualitative description of the synthesis of GO, and **this synthetic procedure is not reproducible as reported!** For example, how much (mL) concentrated H₂SO₄ was used for the synthesis? how the authors prepared graphite-acid solution? what was the amount of graphite used per volume of concentrated H₂SO₄? etc. Also, the authors mentioned in Introduction that they used a modified Hummers' method for the synthesis; **refer to the related paper(s) used for the synthesis in this section.**
 - Lines 133-134:** If TGA was performed under air or inert atmosphere (i.e., N₂)? Add this information to these lines.
 - Line 151:** Change "nutrient agar plate" to "nutrient agar (i.e., NA) plate"
- 4. Results and Discussion:**
 - Line 205:** Start Results and Discussion section with Scheme 1 and discuss it in a few sentences here to give a general overview of the work
 - Line 208:** Change "p-p* transition" to "π-π* transition"
 - Line 209:** Change "n-p* transition of carboxyl" to "n-π* transition of carbonyl"
 - Line 210:** Refer to Figure 2 at the end of the sentence in Line 210
 - Lines 219-221:** Authors refer to Figure 3a and 3b in these lines, but there is no a and b labels in Figure 3 (page 30 of 39); **the same inconsistency is also observed in Figures 4, 5, and 6.**
 - Lines 227-228:** Change "range of 1500-1700 cm⁻¹" to "range of 1300-1700 cm⁻¹"
 - Lines 230-232:** Authors stated that "*The characteristic D peak is not observed in pristine graphite rather it only displays the G peak [40]. In the current study, a sharp graphitic G*

peak is visualized at 1581 cm^{-1} (Figure 4a).”; I also can see a tiny D peak around 1350 cm^{-1} and a 2D peak around 2800 cm^{-1} in Raman spectrum of graphite in Figure 4, which is not discussed in the manuscript.

Lines 237-238: The authors stated that “*FTIR is an important vibrational spectroscopy tool to characterize nanoscale materials.*”; **I do not agree with this statement!** FTIR is a characterization technique, which gives information about functional groups in all types of materials, and it is not specific to “*nanoscale materials*”!

Lines 239-240: Change “ 1610 cm^{-1} (the structural vibration) and 3450 cm^{-1} (the vibration of O-H stretching)” to “ 1610 cm^{-1} (C=C stretching vibration) and 3450 cm^{-1} (O-H stretching vibration)”

Line 243: Change “carboxyl group (–C=O)” to “carbonyl group (–C=O)”

Lines 272-274: What I see in Figure 8 is a smooth surface for graphite and a rough surface for GO; this Figure can be moved to SI.

Lines 307-309: What are those abbreviations in Figure 11 and Table 1? Define those abbreviations or add abbreviations in front of their respective antibiotics in Lines 307-309; for example, gentamycin (i.e., GEN)

Line 314-315: The authors can refer to Scheme 1 at the end of sentence in line 315

Line 322-323: The authors can also refer to Scheme 1 at the end of sentence in line 323

5. **References:** There is inconsistency in reporting references; review and revise accordingly.

6. **Figure Legends, Tables, and Figures:**

Figure Legends: There is inconsistency in Figure Legends and Figures 3, 4, 5, and 6.

Table 2: No need for Table 2 in the manuscript; this can be moved to SI.

Figure 1 and 2: There is no reference to Figures 1 and 2 in the manuscript; refer to these Figures in appropriate areas.

Figures 3, 4, 6, and 7: What X and Y axes represent in Figures 3, 4, 6, and 7?

Figure 5: Change “Wave number (Cm^{-1})” to “Wavenumber (cm^{-1})” in Figure 5; also, what is the peak $\sim 2400\text{ cm}^{-1}$ in FTIR spectra of graphite and GO? Discuss this in the manuscript.

Figure 7: Numbers/elements in Figure 7 are not readable; this Figure can be moved to SI.

Figures 8 and 9: The scale bar and numbers in Figures 8 and 9 are not readable; this Figures need further processing with ImageJ software.

Figure 10: Some information in Figure 10 is not labeled correctly and is not readable.

Figure 11: I see two AZM but no GEM in Figure 11! (for example, check out Figures 11a and b against *E. Coli*); correct this Figure according to Antibacterial Activity discussion.

Figure 13: What is GO concentration in lanes 1-11? Revise Figure Legend accordingly.

Overall, this work is appropriate for publication in RSOS, but there are some grammatical errors throughout the manuscript, inconsistency in Figures and Figure Legends, inconsistency in reporting references, and missing information in Figures or Figure Legends that must be addressed before publication.

Appendix B

Authors responses to reviewer-1

Comments to the Author(s) 10 million people every year by 2050. with (page 3, row 35) -----

[Recommendation-1: Please omit the dot (.) after 2050]

Response- Dot is omitted.

350 ml of deionized water (DI). Than 30% (page 5, row 118) -----

[Recommendation-2: Please write 'Then' instead of 'Than']/

Response- Done accordingly.

the pH of the material was at 6. (page 6, row 122) -----

[Recommendation-3: write the value as '6.0' instead of 6 only, though the value same]/

Response- Done accordingly.

20 µl of prestoblue was added in each well and (page 8, row 186) -----

[Recommendation-4: start with words, not with numerical figure]/

Response- Done accordingly.

used to treat the DNA (10 µl) at 37°C for 2 h. (page 8, row 195) -----

[Recommendation-5: if possible, please mention the concentration of DNA (optional)]/

Response- Thanks reviewer 1 for this wonderful idea! In this study we just focused on the concentration of GO against DNA degradation. To find the exact concentration of DNA working solution loaded in each lane, we need to use Nanodrop. In our following work we have decided to apply this interesting idea.

10 µl of GO treated DNA was mixed with 2 µl of... (page 8, row 195) -----

[Recommendation-6: start with words, not with numerical figure]/

Response- Done accordingly.

Thus it can be assumed that the (page 12, row 331) -----

[Recommendation-7: Please put a comma just after 'Thus']/

Response- Done accordingly.

Thus bacterial strains cannot regenerate the ruptured membrane and (page 13,row 349) -----

[Recommendation-8: Please put a comma just after 'Thus']/

Response- Done accordingly.

For fig. 3 and 4 (Page 30, 31) -----

[Recommendation-9: if appropriate, Label the axis (X- and Y)]/

Response- Done accordingly.

For fig. 6 (Page 33) -----

[Recommendation-10: if appropriate, Label the axis (X- and Y)]

Response- Done accordingly.

Appendix C

Reviewer-2

1. Abstract (Lines 23-26): The authors stated, “The main focus of the antimicrobial activity analysis is... proteins or other biomolecules; this sentence is not necessary in Abstract and can be moved to Antibacterial Activity discussion section.

Response to Abstract: Done accordingly.

2. Introduction:

(i) Lines 48-50: Add proper references at the end of this sentence: In recent years, considerable research has been focused on the bottom-up development of ... and prolonged lifetime. ^x

(ii) Line 78: The authors stated, “Herein, we synthesized GO using a modified Hummers’ method ”; which modification of Hummers’ method exactly? Add a proper reference.

(iii) Line 79: Correct NO₃ to NO₃⁻ - Line 98: Scheme 1 is mentioned here, but not discussed! I recommend the authors to move Scheme 1 to the beginning of Results and Discussion section and discuss it there

Responses to introduction:

(i). A proper reference is added. Please check line no : 52

(ii). The reference “An improved Hummers method for eco-friendly synthesis of graphene oxide” is added. Please check line no: 80

(iii). NO₃ is corrected to NO₃⁻, Please check line no: 81. Scheme 1 is referred to the end of some lines in result and discussion section as suggested later. Please check line no: 322 and 328

3. Materials and Methods:

- (i). Lines 114-123: The authors provided a qualitative description of the synthesis of GO, and this synthetic procedure is not reproducible as reported! For example, how much (mL) concentrated H_2SO_4 was used for the synthesis? How the authors prepared graphite-acid solution? What was the amount of graphite used per volume of concentrated H_2SO_4 ? etc. Also, the authors mentioned in Introduction that they used a modified Hummers' method for the synthesis; refer to the related paper(s) used for the synthesis in this section.
- (ii). Lines 133-134: If TGA was performed under air or inert atmosphere (i.e., N_2)? Add this information to these lines.
- (iii). Line 151: Change “nutrient agar plate” to “nutrient agar (i.e., NA) plate”

Responses to Materials and Methods:

- (i). All the suggestions are added accordingly. Please check line no: 116-118
- (ii). TGA was performed N_2 atmosphere. Please check line no: 137
- (iii). Changed accordingly, please check line no: 154

4. Results and Discussion:

- (i) Line 205: Start Results and Discussion section with Scheme 1 and discuss it in a few sentences here to give a general overview of the work
- (ii) Line 208: Change “p-p* transition” to “ π - π * transition”
- (iii) Line 209: Change “n-p* transition of carboxyl” to “n- π * transition of carbonyl”
- (iv) Line 210: Refer to Figure 2 at the end of the sentence

(v) In Line 210 Lines 219-221: Authors refer to Figure 3a and 3b in these lines, but there is no a and b labels in Figure 3 (page 30 of 39); the same inconsistency is also observed in Figures 4, 5, and 6. Lines 227-228: Change “range of 1500-1700 cm^{-1} ” to “range of 1300-1700 cm^{-1} ”

(vi) Lines 230-232: Authors stated that “The characteristic D peak is not observed in pristine graphite rather it only displays the G peak [40]. In the current study, a sharp graphitic G peak is visualized at 1581 cm^{-1} (Figure 4a).”; I also can see a tiny D peak around 1350 cm^{-1} and a 2D peak around 2800 cm^{-1} in Raman spectrum of graphite in Figure 4, which is not discussed in the manuscript.

(vii) Lines 237-238: The authors stated that “FTIR is an important vibrational spectroscopy tool to characterize nanoscale materials.”; I do not agree with this statement! FTIR is a characterization technique, which gives information about functional groups in all types of materials, and it is not specific to “nanoscale materials”!

(viii) Lines 239-240: Change “1610 cm^{-1} (the structural vibration) and 3450 cm^{-1} (the vibration of O-H stretching)” to “1610 cm^{-1} (C=C stretching vibration) and 3450 cm^{-1} (O–H stretching vibration)”

(ix) Line 243: Change “carboxyl group ($-\text{C}=\text{O}$)” to “carbonyl group ($-\text{C}=\text{O}$)”

(x) Lines 272-274: What I see in Figure 8 is a smooth surface for graphite and a rough surface for GO; this Figure can be moved to SI.

(xi) Lines 307-309: What are those abbreviations in Figure 11 and Table 1? Define those abbreviations or add abbreviations in front of their respective antibiotics in Lines 307-309; for example, gentamycin (i.e., GEN)

(xii) Line 314-315: The authors can refer to Scheme 1 at the end of sentence in line 315 Line 322-323: The authors can also refer to Scheme 1 at the end of sentence in line 323

Responses to Results and Discussions:

- (i). In comment no xii reviewer suggested that, “The authors can refer to Scheme 1 at the end of sentence in line 315 Line 322-323: The authors can also refer to Scheme 1 at the end of sentence in line 323”.Scheme 1 is referred to the end of the recommended lines in result and discussion although the line number is changed. Please check line no: 322 and 328. If the scheme 1 is discussed at the beginning of results and discussion as mentioned in comment no: (i) some information of introduction will be repeated.
- (ii). The change is done. Please check line no: 210.
- (iii). The change is done. Please check line no: 211-212.
- (iv). Figure 2 is referred to the end of the sentence. Please check line no: 210.
- (v). All the missing figure sub number (a and b) are added. Here, **“a”** refers to graphite and **“b”** refers to GO for all the spectroscopy figures except UV-vis.
- (vi). The D peak and 2D peak of graphite is discussed. . Please check line no: 229-236.
- (vii). The statement is removed. Please check line no: 240-242
- (viii). The change is done. Please check line no: 243-244
- (ix). The change is done. Please check line no: 247
- (x). The figure is moved to SI as supplementary figure
- (xi). Changed accordingly, please check line no: 315-317
- (xii). Scheme 1 is referred. Please Check line no: 322 and 328

5. References: There is inconsistency in reporting references; review and revise accordingly.

Response: All the inconsistency is recovered. In previous submission, the multiple references at the same location seemed like [1], [2], [3]. This inconsistency is changed to [1-3], i.e., all the references cited at the end of a sentences are taken inside one bracket.

6. Figure Legends, Tables, and Figures:

(i).Figure Legends: There is inconsistency in Figure Legends and Figures 3, 4, 5, and 6.

(ii).Table 2: No need for Table 2 in the manuscript; this can be moved to SI.

(iii). Figure 1 and 2: There is no reference to Figures 1 and 2 in the manuscript; refer to these Figures in appropriate areas.

(iv). Figures 3, 4, 6, and 7: What X and Y axes represent in Figures 3, 4, 6, and 7?

(v).Figure 5: Change “Wave number (Cm-1)” to “Wave number (cm^{-1})” in Figure 5; also, what is the peak $\sim 2400 \text{ cm}^{-1}$ in FTIR spectra of graphite and GO? Discuss this in the manuscript.

(vi).Figure 7: Numbers/elements in Figure 7 are not readable; this Figure can be moved to SI.

(vii).Figures 8 and 9: The scale bar and numbers in Figures 8 and 9 are not readable; this Figures need further processing with ImageJ software.

(viii).Figure 10: Some information in Figure 10 is not labeled correctly and is not readable.

(ix).Figure 11: I see two AZM but no GEM in Figure 11! (for example, check out Figures 11a and b against E. Coli); correct this Figure according to Antibacterial Activity discussion.

(x).Figure 13: What is GO concentration in lanes 1-11? Revise Figure Legend accordingly.

Responses to Figure Legends, Tables, and Figures:

- (i). There were some inconsistency in legends nomenclature. All are corrected accordingly.
- (ii). Table 2 is moved to SI as supplementary figure
- (iii). Figure 1 and 2 is referred to the end of line no: 122 and 210 respectively.
- (iv). Figure titles are added to all of the figures.
- (v). The change is done. The 2400 cm^{-1} peak is also discussed. Please check line no 254-256
- (vi). It is moved to SI as supplementary figure with clear appearance.
- (vii). The scale bars of FESEM and TEM are now well visualized. Please check Figure no 7 (In Previous manuscript the number of this Figure was 9. This change occurred after sending some figures in the supporting information).
- (viii). The AFM image is processed well and labeled correctly. Please check Figure no 8. (In Previous manuscript the number of this Figure was 9).
- (ix) Changed accordingly, please check figure 9 (In previous manuscript the number of this figure was 10).
- (x) Changed accordingly, please check line no: 679-685